# Isolation of Extracellular Vesicles of Holothuria (Sea Cucumber *Eupentacta fraudatrix*)

**DOI:** 10.3390/ijms241612907

**Published:** 2023-08-17

**Authors:** Anastasiya V. Tupitsyna, Alina E. Grigorieva, Svetlana E. Soboleva, Nadezhda A. Maltseva, Sergey E. Sedykh, Julia Poletaeva, Pavel S. Dmitrenok, Elena I. Ryabchikova, Georgy A. Nevinsky

**Affiliations:** 1Siberian Division of Russian Academy of Sciences, Institute of Chemical Biology and Fundamental Medicine, Lavrentiev Ave. 8, 630090 Novosibirsk, Russia; 2G.B. Elyakov Pacific Institute of Bioorganic Chemistry, Far Eastern Division of Russian Academy of Sciences, 159 100 let Vladivostoku Ave., 690022 Vladivostok, Russia; paveldmt@piboc.dvo.ru

**Keywords:** holothuria *Eupentacta fraudatrix*, late endosomes, exosomes, extracellular vesicles, isolation

## Abstract

Extracellular vesicles (EVs), carriers of molecular signals, are considered a critical link in maintaining homeostasis in mammals. Currently, there is growing interest in studying the role of EVs, including exosomes (subpopulation of EVs), in animals of other evolutionary levels, including marine invertebrates. We have studied the possibility of obtaining appropriate preparations of EVs from whole-body extract of holothuria *Eupentacta fraudatrix* using a standard combination of centrifugation and ultracentrifugation. However, the preparations were heavily polluted, which did not allow us to conclude that they contained vesicles. Subsequent purification by FLX gel filtration significantly reduced the pollution but did not increase vesicle concentration to a necessary level. To detect EVs presence in the body of holothurians, we used transmission electron microscopy of ultrathin sections. Late endosomes, producing the exosomes, were found in the cells of the coelom epithelium covering the gonad, digestive tube and respiratory tree, as well as in the parenchyma cells of these organs. The study of purified homogenates of these organs revealed vesicles (30–100 nm) morphologically corresponding to exosomes. Thus, we can say for sure that holothurian cells produce EVs including exosomes, which can be isolated from homogenates of visceral organs.

## 1. Introduction

The importance of extracellular vesicles (EVs), including exosomes, in mammalian organisms is undoubted, although discussions about the molecular mechanisms of their functioning, molecular composition, and isolation methods will obviously continue for many years to come. The efficient transmission of molecular signals from cell to cell is considered a fundamentally important function of EVs [1,2,3,4,5]. The universality of the established functions of EVs in the implementation of various processes in mammalian cellular systems encourages the study of EVs in organisms at other evolutionary levels, in particular, marine invertebrates. These studies could help understand the mechanisms behind the unique capabilities of marine invertebrates, such as the ability to regenerate efficiently.

Few studies reporting the presence of EVs in marine invertebrates were recently published. Marine life is used in oriental medicine, and studies have recently been published aimed at identifying the “healing” properties of EVs. Thus, EVs isolated from the extracellular matrix of the body wall of the holothurian *Stichopus japonicus* showed a more effective suppression of LPS-induced expression of inflammatory cytokines by macrophages than similar tissue vesicles from the flounder Paralichthys olivaceus and shell-less shrimp Litopenaeus vannamei [6,7]. Such studies undoubtedly contribute to the understanding of the curative effects of preparations derived from marine invertebrates, as well as the substances that enter when these animals are eaten.

The term “extracellular vesicles” comprises vesicles of various sizes and origins circulating in the biological fluids of mammals, among which exosomes are of the greatest interest to researchers. Exosomes (40–100 nm) are formed from the membrane of late endosomes (multivesicular bodies) by budding, accumulate in the endosome lumen, and are called “intraluminal vesicles” before they leave the cell by fusion of the endosome membrane with plasmalemma [1,2,3,8,9,10]. The exosomes were isolated from various mammalian biological fluids, including urine, blood, breast milk, saliva, tears, and amniotic fluid [1,11,12,13,14,15]. Molecules of mRNA, microRNA, proteins, and lipids were found in exosomes, which gave reason to suggest that exosomes are involved in intercellular communications: their components can be transferred to recipient cells, affecting their functions [4,5,8,9,16].

Marine invertebrates, holothurians, are an attractive object for EVs research, as they possess not only nutritional and healing value, but also unique regeneration abilities, which are the “usual” adaptive mechanism for these animals [17,18]. Many holothurians (class Holothuroidea, Phylum Echinodermata) are capable of complete restoration of lost body parts within several weeks after a wide variety of injuries including evisceration or scission into two parts [17,19,20,21]. The mechanisms of regeneration in holothurians are unclear; however, based on current knowledge about the regulation of life processes in mammals, they must be multicomponent. Published studies on the mechanisms of regeneration mainly concern molecular aspects, gene expression, and changes in tissues and organs [10,17,22,23,24]; while the possible role of EVs in the implementation of the regeneration process has not been studied. To fill the gap, we first decided to determine whether exosomes and other EVs are present in holothuria *Eupentacta fraudatrix* and could be isolated.

The most widely used method for exosome isolation is a combination of centrifugation and ultracentrifugation, and numerous varieties of this approach have been published [1,2,5,25,26,27]; however, all the preparation results contain, along with exosomes, more or less different impurities. For example, a study of more than 200 preparations of various biological fluids by transmission electron microscopy (TEM) using specific antibodies to CD63 and CD9 (exosome markers), revealed “typical” exosomes, structureless impurities and spherical particles of low electron density (20–100 nm) without a boundary membrane (“non-vesicles”) in each preparation [14]. The specific morphology of “non-vesicle” lets us classify them as intermediate and low-density lipoproteins (20–40 nm) as well as very low density (40–100 nm). TEM examination of exosome preparations of the female placenta [15,28,29] and mare’s milk [30,31], isolated by centrifugation and ultracentrifugation, revealed all types of impurities described in [14]. Another proof of the need for additional purification of exosome preparations after ultracentrifugation is the detection of up to several thousand different proteins in these preparations [11,31,32,33,34,35,36,37,38,39,40].

Previously, we applied gel filtration for additional purification of exosome preparations obtained from different biological fluids by ultracentrifugation [15,28,29,30,31]. The fact of purification was confirmed by the separation of the exosome peak from two peaks of impurity proteins and their complexes. TEM analysis of the sample obtained after gel filtration showed the presence of vesicles corresponding to “typical” exosomes in terms of morphology and size (30–100 nm), as well as the presence of exosome markers (tetraspanins CD9, CD63, and CD81); the preparations did not contain impurities [15,28,29,30,31]. Thus, we showed the possibility of purification of exosomes obtained from completely different sources [15,28,29,30,31] and decided to apply this technique to analyze the possible presence of exosomes in extracts of the holothurian whole body.

Extracellular vesicles circulating in biological fluids can be uniquely identified as exosomes only when molecular markers CD63, CD81, and CD9 are detected in their membrane with specific antibodies labeled with gold, which bind to vesicles adsorbed on TEM-grids [1,26,27]. Nevertheless, the shape and size (40–100 nm) are the starting point when searching for exosomes, and TEM allows evaluating the presence of vesicles with noted characteristics before proceeding to a rather sophisticated identification of exosomes using labeled antibodies. Now we are at the initial stage of holothuria studies, the use of labeled antibodies is not planned; so, to refer to vesicles whose morphology corresponds to mammalian exosomes, the term “exosome-like vesicles” will be applied. Exosome-like vesicles (ELVs) will be identified by size 30–100 nm, regular spherical or oval shape, and a smooth membrane envelope.

In this work, we analyzed the ELVs of holothurians (sea cucumber *Eupentacta fraudatrix*). They live in a wide range of depths, in all regions of the world’s oceans [20,21]. TEM found extremely rare ELVs in the preparations isolated from holothurian whole-body extracts and purified by gel filtration. We supposed that the concentration of ELVs in these preparations might not be sufficient for detection by TEM.

To understand whether exosomes exist in holothurians, we examined ultrathin sections of holothurian organs in TEM and found late endosomes containing intraluminal vesicles (exosome precursors) in coelom epithelium and parenchyma of the gonads, intestines, and lungs. The vesicles, corresponding to ELVs by structure and size were found in purified homogenates of these organs.

## 2. Results

### 2.1. Holothurian Whole-Body Extract Characteristics

The preparations (sample I) obtained by several standard centrifugation and ultracentrifugation of holothurian whole-body extracts were passed through filters (0.1 μm) and then subjected to gel filtration on Sepharose 4B. A typical profile of gel filtration supernatants is shown in Figure 1. The position of the first peak corresponds to the standard exit site of mammalian exosome elution during chromatography on Sepharose 4B [15,28,29,30,31]. Thus, we obtained the preparation, which could contain exosomes and impurities (sample II).

### 2.2. Electron Microscopic Analysis

First, we tried to isolate EVs from the extract of the whole holothurian body using a combination of centrifugation and ultracentrifugation, this method is considered “a gold standard” for exosome isolation [1,5,27]. The obtained samples (sample I), were negatively stained and examined in TEM (Figure 2).

All samples collected after ultracentrifugation contained huge deposits of structureless material of various electron densities, reflecting the presence of plenty of impurities. Many spherical electron-dense particles and irregularly shaped particles (Figure 2A) were observed over the grid’s entire surface. A feature of the samples (sample I) was numerous “twisted” ribbon-like structures formed by amorphous or granular material of medium-electron density (Figure 2B), which were not detected in mammalian preparations [14]. Membrane fragments arranged singly or in groups (Figure 2C) were present in sample I, as well as spherical particles (25–250 nm) and “non-vesicles” (20–80 nm) of medium electron density and clear or uneven edges (Figure 2A,D). Extremely rare vesicles were found (Figure 2E), their scanty number did not allow us to study their characteristics and identify them as ELVs. Two obvious reasons can explain this negative result: (1) the heavy contamination present in sample I and masking the vesicles, and (2) the low concentration of vesicles. We subjected the samples (sample I) to additional purification using gel filtration and obtained more purified samples (sample II).

Gel filtration significantly reduced the amount of structureless pollution covering the grids in the case of sample I. All the structures in sample II looked more distinct, many small particles of various shapes that could not be identified became visible (Figure 3). Most probably, these particles were derived from cellular debris. Membrane fragments in sample II were noticeably rarer, while ribbon-like structures were much more common than in sample I (Figure 2A and Figure 3A). The material composing the ribbon-like structures had an amorphous or fine-grained structure and a low electron density.

Many irregularly shaped membrane-bordered structures filled with fine granular material were observed in sample II (Figure 3B), such structures were not detected in sample I. Additional “new” findings were distinct non-vesicles and elongated particles composed of two closely adjacent membranes (Figure 3D). Purification of sample I by gel filtration significantly improved its quality; however, it did not reveal the desired vesicles that could be identified as ELVs. Single-rounded vesicular structures were observed in sample II (Figure 3C); however, their morphology does not allow us to identify them as ELVs.

Purification by gel filtration did not reveal ELVs in sample II but made it clear that pollution was not the cause of the negative result, which appears to be due to the low concentration of ELVs in the holothurian whole-body extracts. We decided to determine the presence of late endosomes (“producers” of exosomes) [1,3,5] in the cells of holothurian visceral organs in order to make a suitable selection for ELV isolation.

Holothurian cells have a complete set of organelle characteristics for eukaryotes [17,41,42,43]. The characteristics and representation of one or another organelle, like in all eukaryotes, depend on the functional state of the cells and organs, as well as the existence conditions of the organism.

The holothurians used in our work live at the bottom of the sea, in salt seawater, and at temperatures +10–18 °C. Prior to sampling for research, we kept holothurians in seawater at a temperature of +4–6 °C, these conditions kept them alive; however, we realized that we were studying the ultrastructure of holothurian cells under “survival conditions” different than natural ones. However, at this stage of research, we needed to check for the presence of late endosomes in holothurian cells, and we examined ultrathin sections of the largest organs: the gonads, digestive tube, and respiratory tree.

Each organ in holothurians is covered with coelomic epithelium, which is inseparable from the organ and so is present in all samples. In the cells of the coelomic epithelium, along its entire length around all organs, spherical membrane-bordered late endosomes containing intraluminal vesicles were observed (Figure 4A–E). Figure 4F represents the final event in the life-cycle of late endosome: fusion with plasmalemma.

The ultrastructure of coelomic epithelium cells corresponded to their participation in the transfer of substances between the body parts of the holothurian. The apical surface of these cells had numerous long outgrowths; there were signs of macropinocytosis and rare pictures of “coated” vesicle formation, reflecting the process of clathrin-dependent endocytosis. Cells of the coelomic epithelium showed a large number of membrane profiles containing the vesicles, some of them could be identified as late endosomes due to the regular spherical shape, size of 500–1000 nm, and the presence of intraluminal vesicles (Figure 4A,B). In the cells of the coelomic epithelium, late endosomes were somewhat more common (1–2 endosomes per section of each cell), than in other organs (0–1 endosome per section of 5–10 cells). This may be due to more active processes of internalization of substances by coelomic cells, the link of which is late endosomes.

The presence of late endosomes in the holothurian visceral organs suggests that the isolation of EVs from their homogenates may be more successful than from extracts of the whole body. We sampled the gut, gonads, and respiratory tree, and prepared pooled homogenates, which were treated in the same way as holothurian whole-body homogenates.

Figure 5 shows the data of gel filtration of the supernatant obtained after centrifugation of a homogenized mixture of intestines, gonads, and lungs of holothurians. The first peak (sample III), corresponding to the place of exit from the column of mammalian exosomes, was divided into seven fractions equal in volume (Figure 5) and each fraction was analyzed by TEM.

Figure 6 shows representative images of structural components of the fractions (peak 1, Figure 5) isolated from supernatant obtained after gel filtration of pooled homogenate of holothurian respiratory tree, digestive tube, and gonads (sample III).

Sample III contained ELVs, vesicles with thick envelopes; non-vesicles; aggregates of small protein particles, and structureless deposits of low- and middle-electron-density material (Figure 6). ELVs (Figure 6, upper row) had a spherical or oval shape, 40–100 nm in size; were bounded by a membrane (about 4 nm in thickness), which visually was very similar to the membrane of mammalian exosomes. The EV filling varied in electron density from very high to middle, obviously, depending on stain penetration through the membrane envelope.

Vesicles with a thick envelope have an irregular rounded shape and homogeneous filling of different electron densities (Figure 6A). The envelope has a thickness of about 6 nm, it looks friable due to well-defined granularity. Perhaps, these structures originated from membrane profiles observed in coelomic epithelial cells on ultrathin sections (Figure 4A,B).

Non-vesicles are compact spherical particles of low electron density 20–120 nm in diameter (Figure 6B), their main feature is the absence of a membrane envelope. Small protein particles of low electron density, irregular shape, 20–40 nm in size, form aggregations that stand out against the general background (Figure 6B, insert). Samples III is polluted with large (up to several µm) deposits of structureless material of middle electron density (Figure 6C), often containing non-vesicles.

A comparison of the samples showed that seven fractions obtained after gel filtration differ from each other by the content of ELVs and other components observed in TEM (Figure 6, Table 1). The maximal content of ELVs and minimal content of impurities were observed in fractions 2 and 3 (Figure 5).

## 3. Discussion

Extracellular vesicles of marine invertebrates have become the object of scientific research relatively recently, and very few appropriate studies have been published to date. EVs, including exosomes, were detected in mud crab hemolymph, and their biological properties were described [44]. Sung-Han Jo and colleagues [6,7] isolated EVs from extracellular matrix obtained from the body wall of holothuria *Stichopus japonicus*; muscle tissue of flat fish *Paralichthys olivaceus* and shell-removed shrimps *Litopenaeus vannamei*, and showed their similarity to mammalian EVs. It is interesting that EVs isolated from holothurians demonstrated significantly higher suppression of LPS-induced expression of inflammatory cytokines and chemokines in macrophages than those obtained from the other two sources [6]. EVs were also isolated from Hydra conditioned medium containing the freshwater cnidarian polyp *Hydra vulgaris* for 4 days [45]. A number of papers have also been published in which the term “extracellular vesicles” is used incorrectly, and we will not mention them.

The object of our study is a small holothuria *Eupentacta fraudatrix*, inhabiting the shallow waters of the Western coast of the Sea of Japan. More than 30 studies of this holothuria have been published, presenting data on biologically active substances isolated from holothurians, and features of its transcriptomes [21,22,43,46,47,48,49].

The mechanisms of evisceration and regeneration in holothurians attract the main attention of researchers; however, they are not fully understood to this day. Taking into account the array of data on the role of EVs in the implementation of life processes in the mammalian body, the need to study holothurian EVs is obvious. To isolate EVs, we applied a routine combination of centrifugation and ultracentrifugation method followed by gel filtration, which provided successful isolation and purification of exosomes from various mammalian liquid and organ samples [1,5,14,27]. However, when using this approach on whole extracts of holothurians, it was not possible to obtain preparations containing a noticeable amount of vesicles, which indicated an insufficient concentration of ES in the primary suspension.

It is well known that exosomes are produced by late endosomes [1,2,3,5,8], so we studied ultrathin sections of holothurian organs in TEM to understand how numerous late endosomes are in different organs. The results obtained clearly showed the presence of late endosomes in the cells of the intestinal tube, gonads, and water lungs, with these “parents” of exosomes most often observed in cells of the coelomic epithelium (Figure 4). The formation of intraluminal vesicles on the membrane surface of late endosomes is well documented using TEM of ultrathin sections [1,2,3,5]. At the same time, the fusion of the membrane of late endosomes and the release of vesicles into the extracellular space is declared, but not demonstrated on direct images. We succeeded to capture this event, which is presented in Figure 4F: a cell of respiratory epithelium is releasing intraluminal vesicles that become exosomes once they enter the extracellular environment. Thus, the cells of the organs of holothurians contain late endosomes, which are producers of exosomes, which are one of the varieties of EVs.

The next stage of our work was an attempt to obtain a more concentrated preparation of holothurian EVs. To obtain this, we subjected a homogenate of the digestive tract, gonads, and watery lungs with a firmly attached coelomic epithelium to the same EVs isolation and purification procedures as the whole-body extract of holothurians. The preparations obtained after the final purification step (gel filtration) contained extracellular vesicles, although not in a high concentration. To increase the concentration of vesicles in the preparation, it is obviously necessary to influence the body of holothurians. In particular, keeping holothurians under conditions identical to natural ones can affect the concentration of vesicles in organ homogenates.

The data obtained from the study of ultrathin sections indicated the presence of clathrin-mediated endocytosis (Figure 4B), which is associated with the formation of late endosomes (Figure 4) and, accordingly, with the production of exosomes. The presence of late endosomes in holothurian cells indicates the presence of exosomes in their body, and the known data on the functions of exosomes indicate the need for further study of their biological functions. A noticeable number of late endosomes, producers of exosomes, in the cells of the coelomic epithelium, is a direct indication of the expediency of studying the presence of EVs and the possibility of their identification in the coelomic fluid.

Perhaps the study of exosomes of holothurians will lead to the discovery of a missing link in understanding the mechanisms of evisceration. The existence of such a link is clearly pointed to in a review published in February 2023 [17], which provides a comprehensive analysis of the known mechanisms of evisceration in holothuroids, the review ends with a conclusion about the importance of the perivisceral coelom for the evisceration process and the need to determine the inducer of this process.

## 4. Materials and Methods

### 4.1. Materials

Most chemicals used for this study were provided by Sigma (St. Louis, MO, USA). Sepharose 4B and Superdex 200 HR 10/30 column (GE Healthcare Life Sciences, Marlborough, MA, USA). Reagents for EM studies were purchased from EMS (Houston, TX, USA). Sea cucumbers *E. fraudatrix* were collected from Peter the Great Bay, Sea of Japan, and kept in seawater at +4–6 °C before sampling. The samples of sea cucumbers were frozen at −40 °C and stored until the experiment. The frozen preparations of sea cucumbers were transported from Vladivostok to Novosibirsk in an aircraft freezer at −40 °C.

### 4.2. Preparation of Holothurian Whole-Body Extracts

Equal parts (45.0 g; together 315 g) of seven whole sea cucumbers *E. fraudatrix* including the body wall were used to obtain the extracts. All pieces of cucumbers before lysis were treated carefully three times with a buffer containing mixture of antibiotics (10,000 U Pen/mL penicillin, 10,000 µg Strep/mL ampicillin, 25 µg Amphotericin B/mL streptomycin; SKU:091674049) from MP bio (Irvine, CA, USA), to inactivate and remove possible bacteria on their surface. Then pieces of whole sea cucumbers were homogenized using a buffer containing 10 mM Tris-HCl, pH 8.0; 1.0 M NaCl, 1.0 mM DTT, 1.0 mM EDTA, and the above antibiotics, in a volume ratio of 1:10. The same method was used to prepare extracts of combined samples of isolated gonads, lungs, and intestines (10 g) from 18 to 20 sea cucumbers.

### 4.3. Obtaining Crude Vesicle Preparations

The extracts containing antibiotics were subjected to sequential centrifugations: twice for 40 min at 10,000× *g* at 4 °C, and once for 2 h at 16,500× *g* at 4 °C (Beckman Coulter Avanti-J-301, rotor JA-30.50Ti; Brea, CA, USA). Then the supernatants were passed through a 0.2 μm pore filter. After the filtration, the supernatants were subjected to ultracentrifugation for 2 h at 100,000× *g* (Beckman Coulter Avanti J-30I, rotor JA-30.50). The precipitate was resuspended using 8 mL of TBS buffer (20 mM Tris HCl pH 7.5, 0.15 M NaCl, and antibiotics), and the mixture was centrifuged at 100,000× *g* (Beckman L8-M, rotor SW-60) in 4 mL tubes, and then the precipitate was resuspended by the former solvent, and centrifuged in the same manner. The resulting pellets (crude preparations) were resuspended and passed through a filter (0.1 μm). The obtained preparations were resuspended in fresh TBS containing antibiotics and were used for further purification to obtain EV preparations and TEM examination. These further preparations will be called “sample(s) I”.

### 4.4. Purification of Vesicle Preparations by Gel Filtration

The suspensions corresponding to sample I were additionally purified using Sepharose 4B efficiently separating proteins with molecular weights of 60–20,000 kDa. The suspensions (0.5 mL) were applied on a column with Sepharose 4B (volume 50 mL) equilibrated in TBS (20 mM Tris HCl pH 7.5, 0.5 M NaCl), using chromatograph GE Akta Purifier fractions (1 mL) eluted by the same were collected. The preparations of vesicles and proteins were monitored by absorbance at 280 nm. Fractions were dialyzed for removing NaCl against 20 mM Tris-HCl, pH 7.5 for 16 h at 4 °C and then used for further analysis, and will be called “sample(s) II”.

To obtain sample III, the standard procedures for the destruction of a mixture of preparations of the intestine, gonad, and lungs were carried out, described in Section 4.2. Then all the additional purification procedures described above for preparation II were carried out. The preparation obtained after gel filtration using Sepharose 4B was named sample III. All experiments were performed under sterile conditions.

### 4.5. Negative Staining for TEM Study

To search for the EVs, all holothurian preparations collected at different stages of purification were applied for 1 min on a copper 200-mesh grid covered by formvar film stabilized with carbon. Then a grid was placed for 5–10 s on a drop of 0.5% water solution of uranyl acetate. At each stage, excess liquid was removed with filter paper. The grids (2–3 for each sample) were examined in a transmission electron microscope Jem1400 (Jeol, Tokyo, Japan), and images were collected using a Veleta digital camera (EM SIS, Muenster, Germany). The measurements were made using the iTEM software version 5.2 (EM SIS, Muenster, Germany).

### 4.6. Ultrastructural Studies of Holothurian Visceral Organs

Holothurians were placed in a container filled with diethyl ether vapors to avoid muscle contraction. Anesthetized holothurians were fixed on a dissecting table with the ventral surface up. A shallow longitudinal incision was made along the entire length of the body between the rows of ambulacral legs. The coelomic fluid flowed out of the body cavity, which made the body soft and opened free access to the internal organs for sampling. The incision was continued to the aquapharyngeal complex at the oral end and the exit of the cloaca at the aboral part of the body. First, the gonads, which occupy most of the body cavity volume in holothurian, were excised. Then the digestive tube was removed and divided into a pharynx with a corolla of tentacles, intestines, and cloaca. Next, the lungs were removed. All organs were washed in 0.9% of NaCl several times to clean them of sand and were fixed in 4% paraformaldehyde solution for 24 h.

Pieces of ~5 × 5 mm in size were cut out from the fixed organs. The samples were postfixed in a 1% osmium tetroxide solution for 3 h, and then dehydrated according to the standard scheme in ethanol and acetone, and were embedded in an epon-araldite mixture to obtain hard blocks.

Ultrathin and semithin sections were prepared on an ultramicrotome EM UC7 (Leica, Wetzlar, Germany) using a diamond knife (Diatome, Nidau, Switzerland). The semithin sections of holothurian samples were stained with Azur II and were examined in a Leica DM 2500 light microscope (Leica, Wetzlar, Germany) to choose an area for ultrathin sectioning. Ultrathin sections (~70 nm thick) were prepared on a Leica EM UC7 ultramicrotome (Leica, Germany), contrasted with solutions of uranyl acetate and lead citrate, and examined in a JEM 1400 electron microscope (JEOL, Tokyo, Japan). Digital images were collected using a Veleta side-mounted camera (EM SIS, Muenster, Germany).

## 5. Conclusions

In this work, for the first time, we applied well-established approaches for isolation of EVs from holothuria *E. fraudatrix*. The generally accepted method of sequential centrifugation and ultracentrifugation, which allows obtaining preparations of mammalian vesicles suitable for research, was unable to isolate vesicles from the extract of whole holothurians. Subsequent purification by gel filtration also did not give the desired result.

The study of ultrathin sections in TEM established the presence of late endosomes in the cells of coelom epithelium and parenchyma of the digestive tube, water lung, and gonad of holothurians, and demonstrated the release of intraluminal vesicles from late endosomes into extracellular space. The use of homogenates of these organs made it possible to obtain preparations containing vesicles by methods used for extracts of whole holothurians.

## Figures and Tables

**Figure 1 ijms-24-12907-f001:**
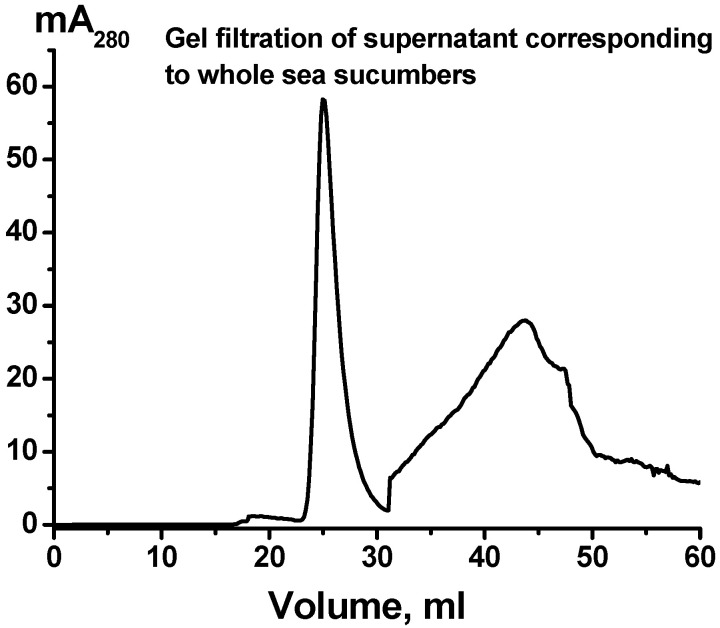
Typical profile of gel filtration of supernatant obtained using whole holothurian body extracts previously partially purified by several different centrifugations (sample I); (^__^), absorbance at 280 nm (mA_280_). The first peak in terms of the elution volume (mL) corresponds to those in the case of “typical” mammalian exosomes with molecular weights 1000 ± 100 kDa [15,28,29,30,31].

**Figure 2 ijms-24-12907-f002:**
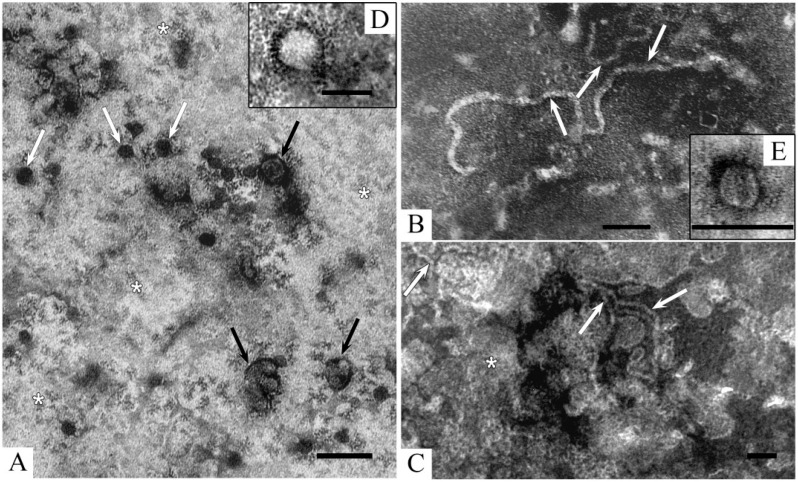
Representative images of structural components in the preparation obtained by centrifugation and ultracentrifugation of holothurian whole-body extract (sample I). (**A**)—general view of the preparation, white arrows show electron-dense spherical particles, black arrows—particles of irregular shape; (**B**)—ribbon-like structures (arrows); (**C**)—membrane-like fragments (arrows); (**D**)—“non-vesicle”, note small “grains” and “chains” around; (**E**)—vesicle (<100 nm). Asterisks show structureless impurities. TEM, negative staining with uranyl acetate. The length of the scale bars corresponds to 100 nm.

**Figure 3 ijms-24-12907-f003:**
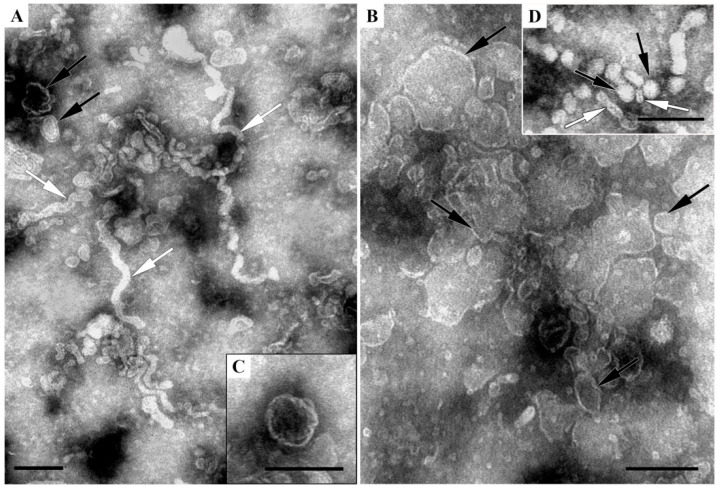
Representative images of structural components in preparation (sample II) obtained by gel filtration of sample I. (**A**)—general view of the preparation, white arrows show ribbon-like structures; black arrows—various membrane profiles; (**B**)—irregularly shaped membrane-bordered structures (arrows) filled with fine granular material; (**C**)—rounded vesicle with uneven edges; (**D**)—spherical non-vesicles (black arrows) and elongated membrane particles (white arrows). TEM, negative staining with uranyl acetate. The length of the scale bars corresponds to 100 nm.

**Figure 4 ijms-24-12907-f004:**
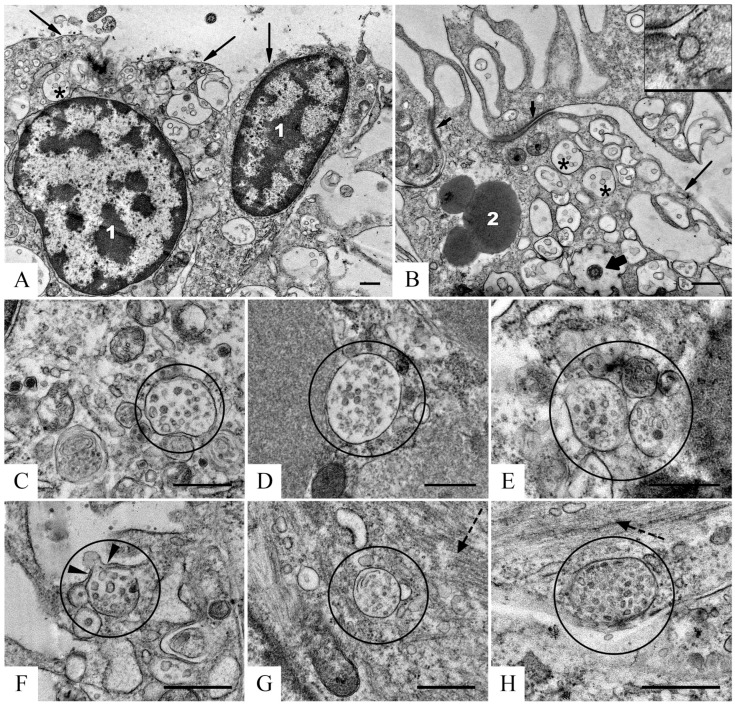
Representative images of holothurian cells. Coelomic epithelial cells covering: (**A**,**B**)—respiratory tree, insert shows formation of coated vesicle; (**C**–**E**)—gonad; (**F**)—cell of respiratory epithelium. (**G**)—myoepithelial cell; (**H**)—smooth muscle cell. Late endosomes containing intraluminal vesicles are shown by asterisks (**A**,**B**) and circles (**C**–**H**). Note fusion of late endosome with plasma membrane evidencing for “a birth of exosomes” in (**F**), arrowheads show sites of contact of two membranes. 1—nucleus; 2—lipid droplet; arrows show apical plasmalemma; short arrows—tight junctions; thick arrow shows cross-section of cilia in scalloped membrane sack; broken arrows show myofibrils. TEM, ultrathin sections. The length of the scale bars corresponds to 500 nm.

**Figure 5 ijms-24-12907-f005:**
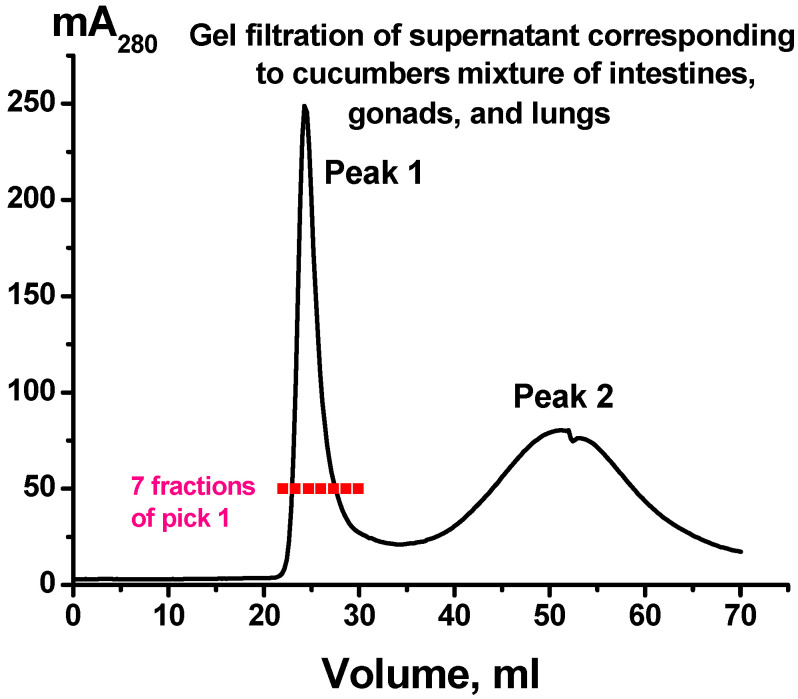
Typical profile of gel filtration of supernatant obtained using mixture of intestines, gonads, and lungs (previously partially purified by several different centrifugations); (^__^), absorbance at 280 nm (mA_280_). Red marks the seven fractions of the first peak. The first peak in terms of the elution volume (mL) corresponds to those in the case of “classical” mammalian exosomes with molecular weights of mammalian exosomes, while second peak—different admixtures [15,28,29,30,31].

**Figure 6 ijms-24-12907-f006:**
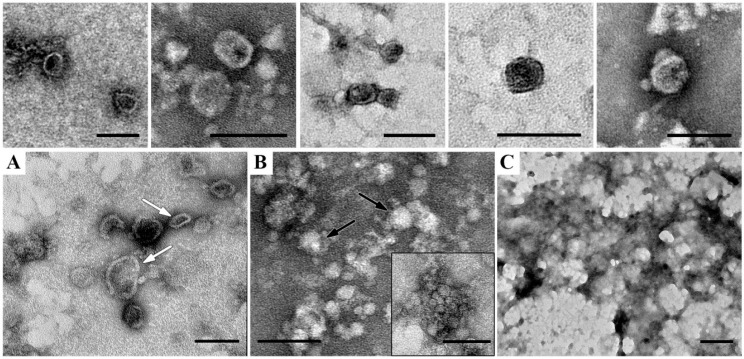
Representative images of structural components in preparation obtained by gel filtration of pooled holothurian respiratory tree, digestive tract, and gonads (sample III), preliminary purified by ultracentrifugation. Upper row—ELVs; (**A**)—vesicles with thick envelope (white arrows); (**B**)—non-vesicles (black arrows) and small particles, insert enclosed an aggregate of small particles; (**C**)—deposits of structureless pollution. TEM, negative staining with uranyl acetate. The length of the scale bars corresponds to 100 nm.

**Table 1 ijms-24-12907-t001:** Evaluation of different structure content in seven fractions of the first peak after gel filtration.

Fraction Number	ELVs	Vesicles with Thick Envelope	Non-Vesicles	Small Protein Aggregates	Deposits of Structureless Material
1	+	No	Single	+	No
2	++	No	Single	+	+
3	++	Single	Single	+	++
4	+	++	++	++	++
5	+	++	++	++	++
6	Single	++	++	++	++
7	Single	+	++	+	+

The presence of structures was evaluated per one mesh of 200-mesh TEM grid. Single: 1–2 structures; +: 1–10; ++: 11–40.

## Data Availability

All data are given in the article.

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
