# Peer review of "Isolation of Extracellular Vesicles of Holothuria (Sea Cucumber Eupentacta fraudatrix)"

_ijms, 2023, doi:10.3390/ijms241612907_

Round 1
Reviewer 1 Report (Previous Reviewer 2)
-Subsections of the results section are not well defined.
-Methods and Materials: The place of collecting sea cucumber and how to transport it to the laboratory should also be mentioned.
Author Response
-Subsections of the results section are not well defined.
-Methods and Materials: The place of collecting sea cucumber and how to transport it to the laboratory should also be mentioned.
Answer:
I was done
Reviewer 2 Report (New Reviewer)
In this manuscript the authors Tupitsyna et al. use conventional centrifugation and filtration methods along with TEM-based morphology in an attempt to demonstrate the existence of extracellular vesicles/exosomes (EVs) in the holothurian species E. fraudatrix. EVs have not been well-studied in marine invertebrates and therefore this study does represent a novel approach and the authors have a track record of publications examining EVs isolated from mammalian organs and secretions. I have the following comments/suggestions:
1) A major problem with the manuscript is the authors claim that they have “obtained unambiguous evidence for the presence of exosomes” in holothurian organs (Line #299). As the authors correctly point out in the Introduction (Lines #89-98), exosomes in mammalian systems are definitely identified only through the use of specific molecular markers, and these are lacking in holothurians. The authors do morphologically identify a few “exosome-like vesicles” (ELVs) in their negative stain TEM images of centrifuged and filtered extracts presented in Figs 2, 3, and 6, however these are far from definitive. In terms of evidence of ELVs from the thin section TEMs of tissues and organs shown in Fig 4, my sense is that the most accurate assessment of this data is that the TEM clearly shows evidence for the presence of late endosomes/multivesicular bodies (MVBs) in these tissues, not evidence necessarily of ELVs. The authors need to tone down their claims of clearly showing the presence of EVs.
2) The Introduction would be improved by providing some additional context for what is known to date about EVs in marine invertebrates, particularly in holothurians. The authors provide some of this in the Discussion (Lines #262-274) but it would be helpful to also provide this in the Introduction.
3) At several points in the manuscript (for example Lines #103 & 147) the authors indicate ELVs in whole body extracts could not be identified because they are “beyond the sensitivity of TEM methods”. I’m not sure what this means, given that the 2 nm resolution of TEM can certainly image ELVs that are 40-100 nm in diameter. Perhaps what is meant is that the low concentration of ELVs make them difficult to detect using a negative stain TEM-based screening method? This language needs to be clarified.
4) In the Discussion the authors indicate that the TEM image in Figure 4F captures a cell of the respiratory epithelium releasing exosomes (Line #297-298), although they make no mention of this in the Results section. I’m not convinced that Figure 4F clearly shows this process and would suggest removing this statement from the Discussion.
5) Additional quantification of certain aspects of the TEM results presented in Figure 4 would be helpful. For example, the authors indicate that late endosomes (MVBs) were “somewhat more common” in the coelomic epithelium - it would be better if the authors provided some quantitative analysis to support this statement. Table 1 presents the evaluation of the structures seen in negative stain TEM of the organ extracts and the legend indicates that “the presence of structures were (sic – should be was) evaluated per one unit of 200 mesh TEM grid” (Line #259). Does this mean that one grid per fraction was counted?
6) The authors refer to their whole body extracts being “polluted” which made it difficult for them to identify ELVs using negative stain TEM (Figure 1). I’m not sure that polluted is the best descriptive term for this, perhaps “contaminated” would be better as the authors state in Line #145.
7) The authors discuss the next stage of their work in the Lines #301-309 focused on organ extracts, however I would argue that a more fruitful approach would be to search for EVs in isolated coelomic fluid using differential centrifugation and filtration methods. The recent review that the authors highlight at the end of their discussion (Ref #15) argues for the presence of an evisceration signaling mechanism in the coelomic fluid and it is possible that EVs play some role in this process.
8) In Lines #143-144 extremely rare vesicles are referred to as being shown in Figure 2C, however I think that this should instead be Figure 2E.
Minor editing of the English is needed.
Author Response
Referee 2
Dear Reviewer, we are grateful to you for your attention to our work and its detailed analysis. We sincerely appreciate your time and efforts. Your comments certainly helped us look at the work from a different angle and improve its presentation.
In this manuscript the authors Tupitsyna et al. use conventional centrifugation and filtration methods along with TEM-based morphology in an attempt to demonstrate the existence of extracellular vesicles/exosomes (EVs) in the holothurian species E. fraudatrix. EVs have not been well-studied in marine invertebrates and therefore this study does represent a novel approach and the authors have a track record of publications examining EVs isolated from mammalian organs and secretions. I have the following comments/suggestions:
- A major problem with the manuscript is the authors claim that they have “obtained unambiguous evidence for the presence of exosomes” in holothurian organs (Line #299). As the authors correctly point out in the Introduction (Lines #89-98), exosomes in mammalian systems are definitely identified only through the use of specific molecular markers, and these are lacking in holothurians. The authors do morphologically identify a few “exosome-like vesicles” (ELVs) in their negative stain TEM images of centrifuged and filtered extracts presented in Figs 2, 3, and 6, however these are far from definitive. In terms of evidence of ELVs from the thin section TEMs of tissues and organs shown in Fig 4, my sense is that the most accurate assessment of this data is that the TEM clearly shows evidence for the presence of late endosomes/multivesicular bodies (MVBs) in these tissues, not evidence necessarily of ELVs. The authors need to tone down their claims of clearly showing the presence of EVs.
Dear Reviewer,
Your comment point that we did not clearly state the criteria for identifying EVs in the text.
The Introduction deals with EVs isolated from biological fluids. Vesicles in liquid preparations are visualized by the method of negative contrasting (the suspension is adsorbed on the EM grid, then contrasted). In this case, exact identification of EVs is possible using specific antibodies labeled with gold. This labeling is a generally accepted criterion for identifying exosomes or other types of vesicles, for example, microvesicles.
Obviously, in the case of holothurian suspensions we cannot apply human-specific antibodies.
Molecular markers of holothurian exosomes have not yet been identified.
The identification of cells and cellular structures is carried out on ultrathin sections, which are completely different from suspension preparations. The identification criterion is the identity of a structure published in the scientific literature. We unambiguously demonstrated the presence of late endosomes (MBT) in holothurian cells and the presence of intraluminal vesicles in their cavity. Since it is known that it is these vesicles that become exosomes when released from the cell, we considered it possible to conclude that the biological fluids of holothurians contain exosomes.
Nevertheless, we agree with you about the need to soften this conclusion, since the vesicles of holothurians detected in suspensions are not identified using the standard criterion, i.e. detection of molecular markers with labeled antibodies.
We have corrected the text in the Introduction and Discussion, we hope this matches your reasoning.
- The Introduction would be improved by providing some additional context for what is known to date about EVs in marine invertebrates, particularly in holothurians. The authors provide some of this in the Discussion (Lines #262-274) but it would be helpful to also provide this in the Introduction.
Published data on EVs in holothurians are indeed scarce. We have placed in the Introduction (page 2, Line3 44-52) the most interesting, in our opinion.
3) At several points in the manuscript (for example Lines #103 & 147) the authors indicate ELVs in whole body extracts could not be identified because they are “beyond the sensitivity of TEM methods”. I’m not sure what this means, given that the 2 nm resolution of TEM can certainly image ELVs that are 40-100 nm in diameter. Perhaps what is meant is that the low concentration of ELVs make them difficult to detect using a negative stain TEM-based screening method? This language needs to be clarified.
In electron microscopy, there is the notion "resolution", it is to this notion the dimensions of 2 nm correspond. The notion "sensitivity" in the application to the method of negative contrast means the content of particles in the preparation. For example, for most viruses, the sensitivity threshold is 105 particles/ml.
We have clarified the phrases where this turnover occurs.
- In the Discussion the authors indicate that the TEM image in Figure 4F captures a cell of the respiratory epithelium releasing exosomes (Line #297-298), although they make no mention of this in the Results section. I’m not convinced that Figure 4F clearly shows this process and would suggest removing this statement from the Discussion.
We believe that Figure 4F demonstrates the fusion of the late endosome with the plasma membrane of the cell. Similar patterns are observed during the fusion of secretory granules, for example, in cells secreting digestive enzymes. The fusion is noted in Legend to Fig.4:
Note fusion of late endosome with plasma membrane evidencing for “a birth of exosomes” in (F).
We extended the Legend and inserted arrows showing contact of late endosome with plasma membrane, and added missed information into the text, Lines 192-193.
- Additional quantification of certain aspects of the TEM results presented in Figure 4 would be helpful. For example, the authors indicate that late endosomes (MVBs) were “somewhat more common” in the coelomic epithelium - it would be better if the authors provided some quantitative analysis to support this statement.
We did not count late endosomes because their number is not significant in this work. We noted that these organelles are more common on sections of the coelomic epithelium (1-2 per section of each cell) than in other cells (0-1 per section of 5-10 cells), and made appropriate changes to the text. Accurate counting of the number of organelles on ultrathin sections of organs requires special sample preparation and is very time consuming; therefore, the estimate we used is appropriate when describing preparations.
Table 1 presents the evaluation of the structures seen in negative stain TEM of the organ extracts and the legend indicates that “the presence of structures were (sic – should be was) evaluated per one unit of 200 mesh TEM grid” (Line #259). Does this mean that one grid per fraction was counted?
No, 2-3 200-mesh grids we examined for each sample (fraction). The expression: “The presence of structures was evaluated per one unit of 200-mesh TEM grid” reflects the average score of the entire sample. This is a kind of semiquantitative analysis, with explanations:
Single: 1-2 structures; +: 1-10; ++: 11-40.
We changed “one unit to one mesh” in the sentence “The presence of structures was evaluated per one unit of 200-mesh TEM grid” for clarity.
We added number of grids for each sample in section 4.4.
6) The authors refer to their whole body extracts being “polluted” which made it difficult for them to identify ELVs using negative stain TEM (Figure 1). I’m not sure that polluted is the best descriptive term for this, perhaps “contaminated” would be better as the authors state in Line #145.
The term "contaminated" is usually used to describe the presence of a virus, mycoplasma, or bacteria in a sample, so we have used "polluted" to emphasize the presence of structureless components.
- The authors discuss the next stage of their work in the Lines #301-309 focused on organ extracts, however I would argue that a more fruitful approach would be to search for EVs in isolated coelomic fluid using differential centrifugation and filtration methods. The recent review that the authors highlight at the end of their discussion (Ref #15) argues for the presence of an evisceration signaling mechanism in the coelomic fluid and it is possible that EVs play some role in this process.
We agree, thanks for your thoughts. We have made appropriate additions to the text.
8) In Lines #143-144 extremely rare vesicles are referred to as being shown in Figure 2C, however I think that this should instead be Figure 2E.
Thank you very much, this is our mistake.
Thanks a lot for the helpful comments.
Professor George A. Nevinsky
This manuscript is a resubmission of an earlier submission. The following is a list of the peer review reports and author responses from that submission.
Round 1
Reviewer 1 Report
Review of the Ms. ijms-2383458-peer-review-v1
Extracellular vesicles of holothuria (sea cucumber Eupentacta 1 fraudatrix)
This study is describing the obtention of Extracellular vesicles (EVs) from the body of holothurians obtained by using a standard combination of centrifugation and ultracentrifugation and subsequent purification by FLX gel filtrationcentrifugation to identy EVs by transmission electron microscopy of thin sections.
Abstract
The abstract section can be improved. Lines 14-2O need some edition, Please delete the paragraph starting “Currently… including invertebrates” in lines 14-15 and describe the improved methodology.
Also add relevant results based on structure and size of ESVs and brief descriptions of the "typical" exosomes, structureless impurities and spherical particles of low electron density.
Introduction
Line 35, change “and first of all” by “like”
In the first part of this section the authors are describing the importance and methodological processes used to obtain EVs. And in the end of the Ms (Lines 95 backwards), the authors describe Exosome-like vesicles (ELVs).
In this study the authors are describing EVs or ELVs?. If this is the case, the authors should indicate it in the title and subsequent sections.
M & M
L328. Preparation of the extract
Please edit this section; the authors are using 40 g of seven organisms (320 g), use this information for the subsequent descriptions. Delete 10 pieces.
The whole organisms used included the body wall?,
L351. Delete “the suspensions obtained in step 2.2”.
Results
L106—108 Delete “obtained by several standard centrifugation and ultra-centrifugation of holothurian whole body extracts were passed through filters (0.1 µm) and then subjected to gel filtration on Sepharose 4B”. This was referred in the M & M section.
L121. The standard centrifugation and ultra-centrifugation methods considered as a gold standard procedure to obtain EVs, should be mention in the introduction and M & M section, to highlight the importance of the implementation of the methodology used in this study.
L133 delete “a plenty”
L142-145. “Two obvious reasons can explain this negative result: (1) the heavy contamination present in sample I and masking the vesicles, and (2) the low concentration of vesicles, which is beyond the sensitivity of TEM. We subjected the samples I to additional purification, gel filtration, and obtained more purified samples II”. The authors are discussing their results in this paragraph. Please delete from this section and add it to the discussion section.
L172-182. “Holothurian cells have a complete set of organelles characteristic for eukaryotes [15,39-41], and, like in all eukaryotes, the characteristics and representation of one or an-other organelle depend on the functional state of the cells and organs, as well as existence conditions. The holothurians used in our work live at the bottom of the sea, in salt sea water and at temperatures (+10 - 18oC). Prior to sampling for research, we kept holothurians in seawater at a temperature of +4 - 6°C, these conditions kept them alive, however, we realized that we were studying the ultrastructure of holothurian cells under “survival conditions” different from natural ones. However, at this stage of research, we needed to check for the presence of late endosomes in holothurian cells, and we examined ultrathin sections of the largest organs: gonads, digestive tube, and respiratory tree. Each organ in holothuria is covered with coelomic epithelium, which is inseparable from the organ and so is present in all samples”.
This paragraph is out of context in this section, please pass it to the discussion section.
L212. Please describe fully in the M & M section the description for obtaining sample III.
L233. ELVs or EVs?,
L233-238. Some information from this paragraph is useful for the abstract section.
Discussion
L261-273. The first paragraph is not contributing to the discussion section of this paper
L312-319. “The results of our study clearly indicate the presence of exosomes in the body of holothurians, and the known data on the functions of exosomes point to the need for further study of their biological functions. Perhaps the study of exosomes of holothurias will lead to the discovery of a missing link in understanding the mechanisms of evisceration. The existence of such a link is clearly pointed in a review published in February 2023 [15], which provides a comprehensive analysis of the known mechanisms of evisceration in holothuroids, the review ends with a conclusion about the importance of the perivisceral coelom for the evisceration process and the need to determine the inducer of this process”.
It is important to address fully this observation. Please edit the paragraph based on your results and the links mentioned in reference 15.
Comment: The discussion presented in the manuscript should be improved. Overall, the main function of the discussion is to answer the research question posed in the introduction and to use the study's results to pose an answer, or the discussion explains what the study results mean and what contributions the paper makes to the area of study. It should focus on explaining and showing an overall evaluation of what you found, how it relates to your literature review and research questions and making an argument supporting your conclusion. I recommend that authors review the discussion carefully, adding elements that meet the high standard of a good paper.
General comments: In the way that this paper is described, I'm not convinced about the innovation of the study. The authors are using standard procedures previously addressed in mammals. The authors should explore the biology of sea cucumber, the complexity of their body wall and main tissues. This organism is full of GAGs, collagen, fibers that could be interfering with the obtaining of EVs or ELVs. Several speculations need to be confirmed. For instance the identification of late endosomes is novel and worth to be studied fully.
The paper needs extensive English revision
Reviewer 2 Report
The topic is interesting and innovative and valuable, but the structure of the results is such that it raises many questions for the reader and needs to be improved.
Results and Discussion
The division of the results is not good, first of all, in section 3.1, the reader expects the results of all analysis related to sample 1 to be given like its TEM. Then, in section 3.2, only TEM images should be presented based on the title, but the rest of the results have been given.
It is better to divide the results section based on three samples, then, discuss and compare them in the discussion section.
Materials and Methods
Line 328,4.2. Preparation of holothurian whole body extracts: Please add the device you used for homogenizing whole body and the time.
Line 352: it seems that “step 2.2” must be changed to “step 4.3”
There is no explanation about sample III preparation in the method and material section.
Reviewer 3 Report
To understand whether ELVs exist in holothurians, authors examined ultrathin sections of holothurian organs, and found late endosomes containing intraluminal vesicles in coelom epithelium and parenchyma of the gonads, intestines, and lungs. The study showed that vesicles, corresponding to ELVs by structure and size were found in purified homogenates of these organs. This is a very interesting study and new finding in the Holothurian. Authors certified the appearing of extracellular vesicles in holothurian Eupentacta fraudatrix, this kind of marine invertevrates. I think this is a pretty good study and finding. Some issues should be clarified before publication as follows:
1. Some information of holothurian Eupentacta fraudatrix should be provided such as location, species, annual output in Russia or Asian or the whole world and so on.
2. Study progress of extracellular vesicles should be reviewed in the part of introduction.
3. In the figure 6B, the photo is not clear enough.
4. I think the line 300-308 were no need.
5. Line 309-319 were also no need.
6. Authors should strengthen the discussion content based on own obtained data, but not too many outlook or expectation.
Reviewer 4 Report
Letter to Authors
ijms-2383458-v1
Extracellular vesicles of holothuria (sea cucumber Eupentacta fraudatrix)
Anastasiya V. Tupitsyna, Alina E. Grigoriev, Svetlana E. Soboleva, Nadezhda A. Maltseva, Sergey E. Sedykh, Julia Poletaeva, Pavel S. Dmitrenok, Elena I. Ryabchikova, Georgy A. Nevinsky
230531
Dear authors,
You have successfully isolated exosomes from a sea cucumber for further characterization. Your MS is worth publishing in the journal you are submitting to. Before publication, however, your MS needs minor revision. Back-and-forth arguments in the introduction section make this MS difficult to see. Extensive English editing is necessary.
See below for detail. Words in braces indicate options. Bracketed words can be omitted.
L4
and -> delete
Do not break lines.
L11
same -> equal
See L404,405,406.
/ Institute of Chemical Biology and Fundamental 11 Medicine, 8 Lavrentiev Avenue, Novosibirsk 630090, Russia (redundant) -> delete
L27 keywords
holothuria Eupentacta fraudatrix; extracellular vesicles -> replace
Avoid listing words which appear also in the title. Duplicate hits upon computer search do not make sense. Give words that do not appear in the title to draw attention from wider readership. Posting words that neither appear in the abstract is better, because even in full-text search/indexing robots may not weigh much on words deeper (posterior) in the text. Hint: Sepharose gel filtration, agarose-dextran composite gel, ultracentrifugaton, etc.
L34
first of all -> above all
mammalian organisms -> {mammals, mammalian organs}
L56
unclear; however -> unclear. However (break sentence here)
L59
organs [8,15,20-22]; while -> organs [8,15,20-22], while
L63-76
Most widely used method .. in these preparations [9,29-38]. -> move to L50 next to moving L87-92
Some revision may be necessary to fit in.
L64
published [1,2,5,23-25], however -> published [1,2,5,23-25]. However
L65
resulted -> {subsequent, resultant}
L74
in -> previously
L78
fact of purification -> purity
L87-92
Extracellular vesicles .. using labeled antibodies. -> move to L50
Some revision may be necessary to fit in.
L102,etc
lung ?
I am not an echinoderm biologist but wonder if it is an air-breezer owing lungs. Gill?
L108
Typical profile .. is shown on Figure 1. -> delete
Wording like "X is shown in figure/table Y" imposes killing readers' times to read such an information deficient sentence telling only that there is a figure/table. You should present an outline or a perspective drawn from the figure/table and cite it in parentheses at the end. In this case it is at the end of next sentence.
L113 figure picture
whole sea sucumbers -> [whole sea cumber] supernatants
L115
What is "(_)"?
L124 figure 2 picture
Asterisks in pale color are too small to see.
L141
2C ? -> 2E ?
L146,168,231
pollution -> impurity
L198
insert ? -> an insert in B ?
Where is an insert in A?
L204
corresponds (grammar) -> correspond
L206
holothuria -> holothurian
L248
polluted -> contaminated
L258 table 1 body
No -> - (minus)
Single -> ±
OR, - , + , ++, +++
Consistent series of symbols is better.
L278-287
This paragraph needs revision, when you are focusing on regeneration. Instead of whole body extracts, use of regenerating fronts with omnipotent cells should be needed.
L328
whole body -> delete
Extracts were not limited from whole body. See L355.
L343
TBS buffer (20 mM Tris HCl pH 7.5, 0.15 M NaCl, and antibiotics) -> TBS buffer (20 mM Tris HCl pH 7.5, 0.15 M NaCl) with antibiotics
Was NaCl concentration different from the buffer mentioned in L354?
L354
TBS (20 mM Tris HCl pH 7.5, 0.5 M NaCl) (redundant) -> TBS ?
L364
filter paper -> a filter paper
L365
Jem1400 -> JEM1400
Jeol -> JEOL
L369
avoid -> prevent
L379
from -> off
a sand (grammar) -> {sand, sand particles}
L387
(Leica, Wetzlar, Germany) (redundant) -> delete
See L384.
L389
(Leica, Germany) (redundant) -> delete
L390
electron microscope (JEOL, Tokyo, Japan) (redundant) -> delete
See L365.
L391
Veleta side-mounted camera (EM SIS, Muenster, Germany) (redundant?) -> Veleta side-mounted camera ?
What difference with the "Velta digital camera" in L366?
L418 references
Check the reference list carefully again from the beginning. Reference lists are frequently hotbeds of errors. You might add, omit or swap citation in the main text on the way internal revision. Numbering of the references might then shift. If so, readers think you are making irrelevant citation. It is the authors' responsibility that all references are properly cited.
Check thoroughly to make sure:
if scientific names are in Italics,
if paper titles are in lower case (L419,etc),
if journal titles are abbreviated when possible (L420,etc),
if book titles are in Italic title case (L452,etc),
etc.
See the citation guide at:
https://www.mdpi.com/authors/references/
Several grammatical errors can be seen especially regarding counting.